# LHW-Net: An ensemble-based machine learning framework for brain tumor classification

Thireesha Suryadevara[1], Naveenkumar Mahamkali[1], Mudassir Rafi[1,2*]

1 Department of Computer Science and Engineering, SRM University AP, Amaravati, Guntur, Andhra Pradesh, India, 2 Department of Computer Science, College of Computer Science, King Khalid University, Abha, Kingdom of Saudi Arabia

☯ These authors contributed equally to this work.
* mudassir.r@srmap.edu.in, mrafi@kku.edu.sa

## Abstract

The classification of brain tumors is an unsolved problem associated with heterogeneity of tumors and fluctuations in imaging conditions. In this work, the investigation introduces a powerful novel framework, named LHW-Net that combines handcrafted features called local binary patterns (LBP), histogram of oriented gradients (HOG), and wavelet transform (WT). Within the LHW-Net framework, the extracted features are utilized in different machine learning classifiers, such as K-Nearest Neighbors (KNN), Random Forest (RF) and Support Vector Classifier (SVC). The results of the individual classifiers are further combined using probabilistic score fusion approach to improve classification performance. The effectiveness and robustness of the proposed work are validated by the achieved experimental results on commonly accepted benchmark datasets.

## Introduction

Brain is an important organ that is at the hub of the human nervous system and collaborates with the spinal cord in order to coordinate body processes. Being a main control center, it receives the sensory information and passes the guidance with the help of complex neural networks which results in appropriate coordination of the body functions [1]. Brain disorders may significantly affect human health and brain tumors caused by abnormal tissue proliferation inside the skull are considered as one of the most dangerous and life-threatening disorders [2]. The World Health Organization (WHO) grades the type of brain tumors based on the level of severity which implies that Grade I to Grade IV brain tumors are ranked as the least to most severe respectively. Both primary and secondary brain tumors are forms of brain tumors but primary brain tumors may be either benign or malignant and they do not spread to other parts of the body but on the other hand secondary brain tumors are malignant only. Grade one tumors like the meningiomas and gliomas are largely non-threatening, whereas grade four tumors are cancerous, malicious and excessively violent [3].

**Data availability statement:** All the data used in the publication are publicly available in repositories with following URL's Brain Tumor MRI Dataset: https://www.kaggle.com/datasets/masoudnickparvar/brain-tumor-mri-dataset. Brain Tumor Image Dataset https://www.kaggle.com/datasets/denizkavi1/brain-tumor/data. Br35H Brain Tumor Detection 2020 https://www.kaggle.com/datasets/ahmedhamada0/brain-tumor-detection.

**Funding:** The authors extend their appreciation to the Deanship of Research and Graduate Studies at King Khalid University for funding this work through Large Research Project under grant number RGP2/588/46.

**Competing interests:** The authors have declared that no competing interests exist.

Benign tumors have a slow growth rate, and neither do they spread to other parts of the body, but malignant tumors are invasive and are dangerous to the health. The brain tumors form some of the most fatal medical conditions that play significant role towards mortality in all age categories such as children, adults, and other older age groups. For the purpose of devising the most preferable plans of treatment, right classification and proper diagnosis of brain tumors are crucial and this, in turn can even enhance the survival of patients. Tumor detection and estimation is commonly done using advanced imaging tools [4] like Magnetic Resonance Imaging (MRI) [5], Single Photon Emission Computed Tomography (SPECT) [6], Positron Emission Tomography (PET) [7] and Computed Tomography (CT) [8]. Magnetic resonance imaging is an exemplary modality among the following, and its greater resolution, high contrast imaging, non-invasiveness, as well as lack of ionizing radiation, makes this extremely important as a diagnostic tool as well as monitoring the brain tumors [9]. Besides this, MRI has been an area of active research along with the development of Machine Learning and deep learning methods where the modality has a significant potential of resulting in a better diagnostic accuracy as well as the help in devising effective treatment plans to aid patients with better prognosis [10].

Despite the amount of research invested in formulating strong and precise procedures through which the automatic classification can be performed, it is still a very difficult task, because the morphology and the texture of the tumor, as well as its contrast in MRI images is, in general, very heterogeneous. As such, it prevents the development of a universal and trusty method that could be helpful in effective classification of tumors. In the endeavor to solve these problems, the current paper presents such an ensemble method where the advantages of various techniques remain united in a single form, thus covering the flaws of each other. The main contributions of the current work are as follows:

- The suggested framework exploits Local Binary Pattern (LBP), Histogram of Oriented Gradient (HOG) and Wavelet Transform to develop a reliable and unique characteristic descriptor.

- It further uses a probabilistic score combination to combine classification scores of multiple machine learning models, hence enhancing reliability.

- The presented fusion strategy is tested on several common standard benchmark brain tumor datasets, both binary and multiclass classification conditions were considered.

- The test results are compared with the modern approaches indicating that the proposed one has much better classification performance.

The paper is organized as follows: In the Related Work section, the review of prior research is carried out in detail. After that, the data used and the suggested LHW-Net are described in the Materials and Methods section. Then, in the Experimental Results and Discussion section, the experimental setup, and results are presented as

well as the findings discussed in terms of detail. Finally, the Conclusion section provides a conclusion to the main findings, insights and contributions that this work has made.

## Related work

The problem of identifying and classifying brain tumors with machine learning and image processing is a complicated issue using handcrafted feature-based approaches and deep learning algorithms. This section reviews key machine learning and deep learning models for brain tumor classification (BTC).

F. Ullah *et al.* proposed an extended evolutionary lightweight model, which is an extension of recently proposed Multimodal Lightweight XGBoost framework. It is a model particularly adapted to the recognition, and classification, of brain cancer [11]. The suggested model is tested on the dataset that is BraTS 2020 and its results are outstanding as the accuracy is 93.0%. A classification method was proposed based on medical images in order to distinguish between brain tumors and the lesions of an autoimmune disease through multi ensemble machine learning. The model utilizes texture-feature ranking, SVM learner and majority voting method to increase the accuracy of classification [12]. It is especially applicable in determining the existence of multiple sclerosis in the patients of glioma or the other way around. It is shown that the model has been successful with experimental results producing consequent training and testing accuracy of 97.96% and 97.74% respectively. One study used MRI images to classify brain tumor types using six machine learning algorithms such as RF, CN2 Rule Induction, Naive Bayes, SVM, Neural Networks and Decision Tree (DT) [13]. A collection of 253 images was used and 2048 features extracted. Support Vector Machine(SVM) showed the best results compared to other algorithms, giving an accuracy of 95.3%, improving the performance of the BTC models significantly. Normalization techniques were used in order to refine MR image quality, after that DSURF and HOG features are extracted, and they both lead to a better performance. A linear kernel SVM has been used as a classification method with the accuracy of 90.27% [14].

Kaplan *et al.* used an idea of detaining brain tumors applied to two strategies of feature extraction, nLBP and $\alpha$LBP. The nLBP approach measures distance relationships between the neighboring pixels, whereas $\alpha$LBP approach measures angular relations between the pixels. Extracted features using such techniques and by traditional LBP were tested with Linear discriminant analysis (LDA), Artificial Neural Network (ANN), K-Nearest Neighbors (KNN) and RF algorithms in a private dataset. The highest accuracy of 95.56% was attained by nLBP with d = 1 combined with the KNN classifier [2]. In [15] a new texture analysis in tumor images was proposed with integration of LBP and Gray Level Co-Occurrence Matrix (GLCM) features. SVM was then used to classify the fused features with great accuracy of 99.84% which proves the efficiency of this approach in tumor classification.

Machine learning classifiers together with the deep feature extraction were used to develop a method for BTC. Brain MRI scans were processed by deep modeling to generate deep features employing transfer learning models based on pre-trained convolutional neural networks. The highest features were selected to form an ensemble and to be classified depending on the different algorithms. Three brain MRI datasets were assessed, and it was found that applying this method led to an improvement in performance by significant margins, the SVM with Radial Basis Function (RBF) kernel saw the best results, especially when dealing with large amounts of information [16]. In case of brain tumor multiclass, a deep feature fusion method was suggested where preprocessing was done using min-max normalization, but due to data scarcity, a minimal gradient ascent and severe data augmentation were adapted. The combination of features of the transfer-learned networks (namely, AlexNet, ResNet18, and GoogLeNet) was incorporated into a single enriched feature vector [17]. This has made this vector classification by SVM and KNN, thus improved classification performance. The technique outperformed the other existing systems where the accuracy was 99.7%. In [18] Wavelet Transform (DWT) is used to extract features of brain MRI images and a classification is then performed with a Convolutional Neural Network (CNN). Through experimental findings, it is found that such an approach is better than the traditional ones and it has a total accuracy of 99.3%.

## Materials and methods

Fig 1 demonstrates the proposed methodology, LHW-Net, to be used as brain MRI image classification. LHW denotes a combination of LBP, HOG, and Wavelet features. It includes preprocessing the dataset, feature extraction, classification of the data by machine learning algorithms, and accumulation of probabilistic scores to come to final predictions. The approach was evaluated in terms of performance metrics, namely accuracy, precision, F1 score and sensitivity. The information concerning each step such as dataset features, preprocessing, feature extraction, classification, and score fusion is provided in detail in the following subsections.

### Datasets

The significance of using brain MRI in recognizing and distinguishing different brain tissues makes BTC a crucial research topic for both healthcare professionals and image processing analysts. Among the many advantages that MRI scans entail, some of them are that they are not invasive, they do not use radiation hence, safe to use, the ability to take images in different directions, and they encourage multi-dimensional analysis [19]. The suggested method is using three publicly available MRI brain databases to classify a brain tumor successfully. The detailed description of the datasets along with download links is provided in S1 File.

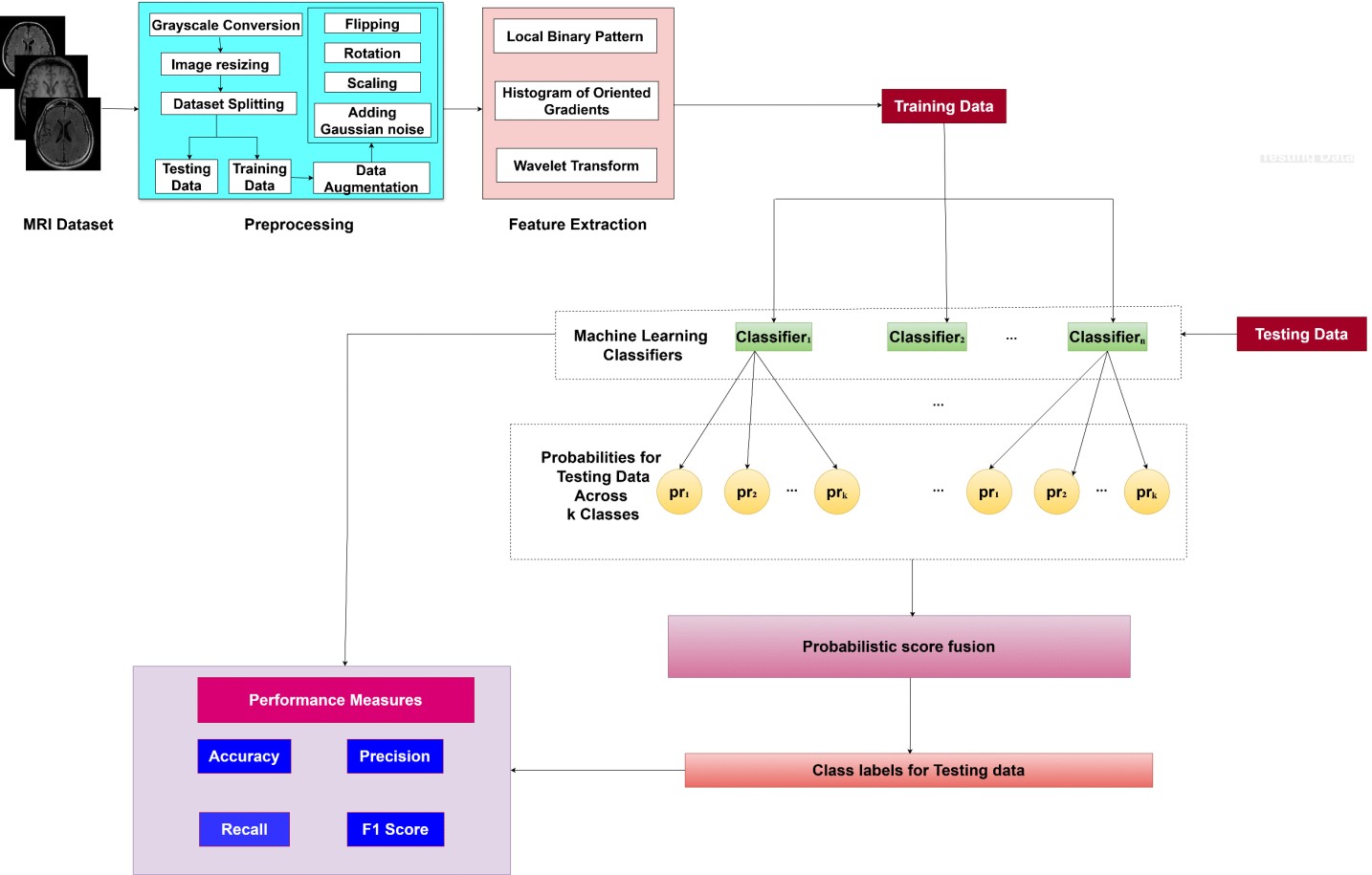

**Fig 1. Detailed architecture of the methodology.**

**Brain tumor MRI dataset.** This data is taken from three sources, Figshare [20], SARTAJ dataset [21] and Br35H dataset [22]. It includes 7,023 magnetic resonance images of human brain divided into four groups: glioma, meningioma, no tumor and pituitary tumor [23]. It should be noted that all the images identified as No Tumor were taken out of the Br35H dataset. The data has been divided into independent training and test sets as shown below:

- **Training set:** 1595 no-tumor, 1457 pituitary, 1339 meningioma, and 1321 glioma images.

- **Testing set:** 300 glioma, 405 no-tumor, 306 meningioma, and 300 pituitary images.

**Brain tumor image dataset.** The brain tumor image dataset consists of 3064 T1-weighted contrast-enhanced MRI scans obtained by 233 patients [24]. It is subdivided into three categories of tumors consisting of meningioma which has 708 images, pituitary tumor having 930 images, and glioma with 1426 images.

**Br35H brain tumor detection 2020.** This dataset comprises 3000 MRI images which are uniformly balanced to two categories as 1500 images are given label of Yes (tumor-positive) and 1500 of label No (tumor-negative) [22].

To make it easy to refer, hereinafter, the three datasets will be known as Dataset I, Dataset II, and Dataset III, that is similar to their respective descriptions in the above subsections. The frequency of the total number of the images per a class over these datasets is shown in Fig 2.

## Preprocessing

Data collection is followed by preprocessing which is a significant process to ensure that the images become more applicable to the intended applications. When preprocessing is done properly, it enhances accuracy, performance of the model, and generates more stable results. The essential operations performed in this research are grayscale conversion, image resizing, partitioning of datasets, and augmentation of data to balance the classes in the training set. First, images were loaded to each class separately to keep appropriate label mapping. To learn the supervised information, labels of classes were transformed into numbers. The Preprocessing steps are outlined as below:

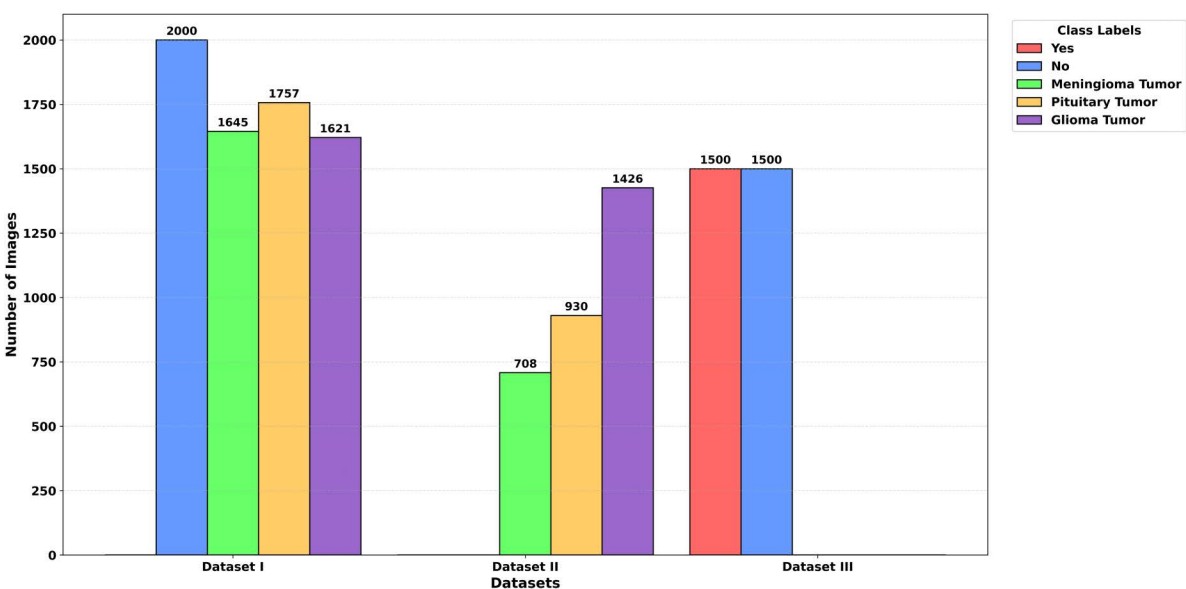

**Fig 2. Summary of brain tumor datasets.**

- **Grayscale conversion:** To reduce the complexity of the data and highlight features according to their intensities, all pictures were converted to grayscale. The images were stored in the original grayscale intensity range, which is in the range of [0, 255].

- **Resizing:** The sizes of the original images constituting each dataset were different (e.g., 512×512, 287×348, 766×879). All images were resized to the size of 128×128 pixels to guarantee consistency in the spatial representation of the images and the extraction of the features.

- **Dataset partitioning:** Where the dataset was not split, the images were split into training and testing at a ratio of 80:20. This provided adequate training data and kept a separate dataset to evaluate as shown in Fig 3.

- **Data augmentation:** The training set was a comparatively small collection of images and unbalanced categories, so augmentation was used to introduce more variability in the data and avoid overfitting. The augmentation techniques were:

  - **Horizontal flipping:** Reflecting the images by flipping horizontally.

  - **Rotation:** Rotation of the images between the range of ±15° in order to mimic various orientations.

  - **Scaling:** To resize the images to create some form of variation in size so that the aspect ratio remains the same.

  - **Addition of Gaussian noise:** To improve invariance to variability, a random amount of noise is added to the images.

Fig 4 shows the various image augmenting techniques. In order to prevent any chances of leakage of data, augmentation was implemented on training data only. The datasets had to be increased to be sure of adequate samples to use in training and testing, and the final values of augmented images available in each dataset are illustrated in Fig 5.

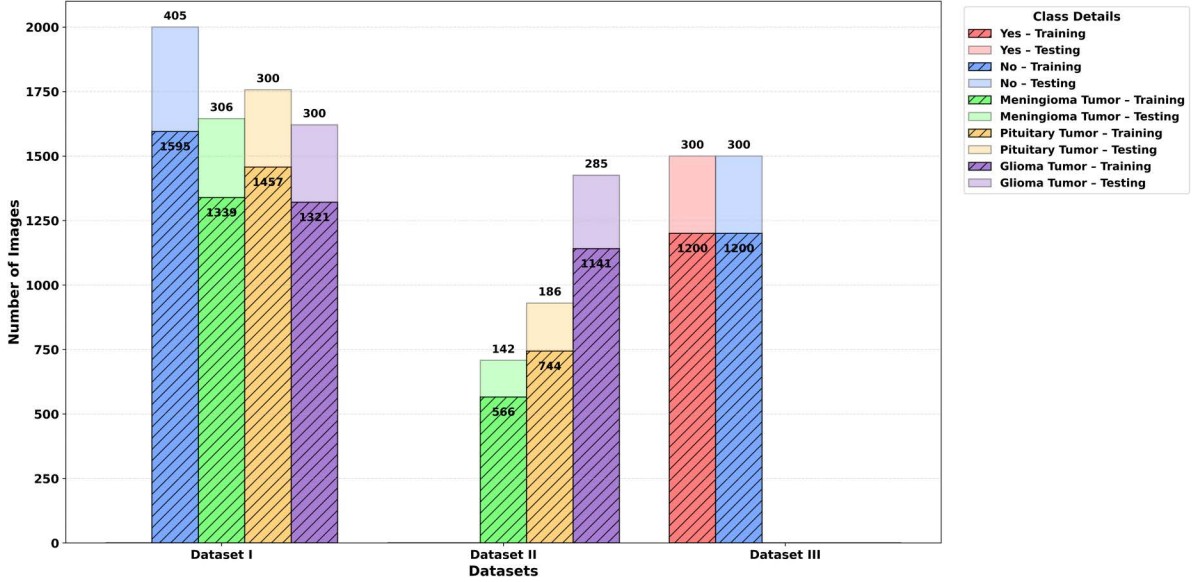

**Fig 3. Overview of brain tumor datasets after training and testing splitting.**

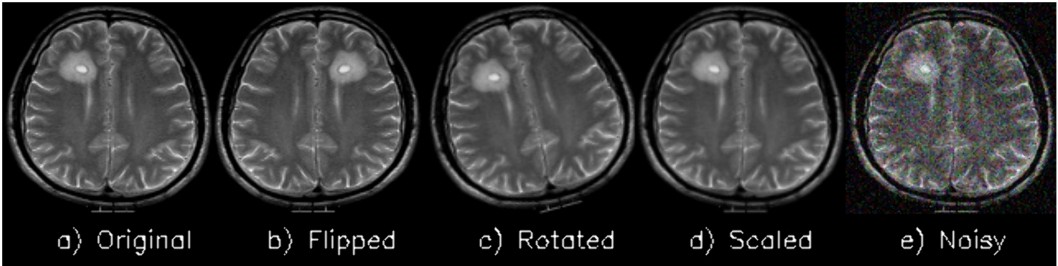

**Fig 4. Visual representation of images after applying various augmentation techniques.**

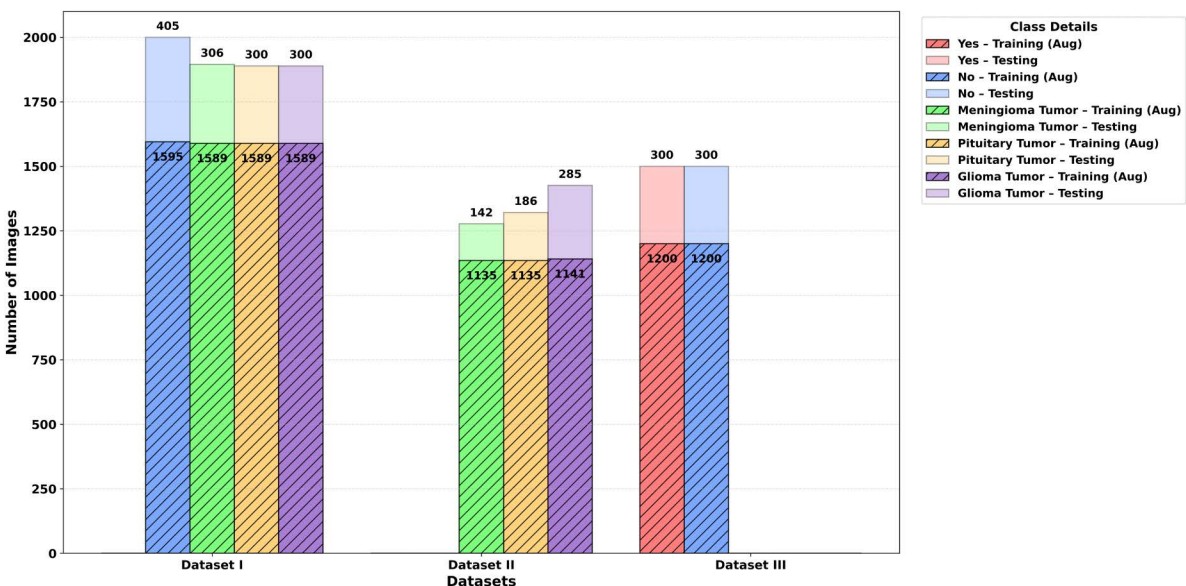

**Fig 5. Overview of brain tumor datasets after augmentation.**

## Feature extraction

Feature extraction is a key stage in machine learning, which converts raw data into humanly interpretable representations by isolating features describing the texture, shape, and intensity. As part of analysis of brain MRI, there is a need to obtain complementary information on the images in order to enhance classification success. In this regard, we use three highly developed descriptors which are Wavelet Transform, LBP and HOG, covering characteristically various physical attributes of the image.

The Wavelet Transform in particular is an effective way of capturing multi-scale changes both in frequency and in intensity, and thus captures both global and local contrast, respectively. LBP considers intensity variations between a pixel and its neighbors to obtain the local texture patterns and thus will be very applicable in obtaining the fine textural details. The HOG technique is interested in the direction of the edges, distribution of the gradients, which encodes the shape and structure of the image.

We make use of the complementarities that exist in these descriptors hence allow us to describe brain MRI images better. In-depth descriptions of every technique of feature extraction follow below. These acquired features are then fused

together using the suggested fusion strategy, compared with machine learning categories and the findings are displayed and examined in Experimental Results and Discussion section.

**Local binary pattern.** LBP is an important texture description which describes local structure of an image by contrasting the intensity of a central pixel with its neighbours. It has already been useful in texture representation, finding usage in fields like facial recognition, texture classification [25]. This study used 8 sampling points ($P = 8$) and a radius of 1 ($R = 1$) in extraction of LBP features with the use of the uniform encoding scheme. Flattening of resulting feature vectors into one dimension scaled arrays was done to be compatible with machine learning models.

The computation of the LBP value for a pixel is expressed as shown in Equation (1):

$$LBP(x_c, y_c) = \sum_{p=0}^{P-1} s\big(g(x_p, y_p) - g(x_c, y_c)\big) \cdot 2^p$$

(1)

where:

- The coordinates of the center pixel are denoted by $(x_c, y_c)$.

- The intensity value at this center pixel is represented by $g(x_c, y_c)$.

- The intensity of the neighboring pixel indexed by $p$ is given as $g(x_p, y_p)$.

- The total number of neighboring pixels considered in a circular arrangement around the center is $P$.

- The index $p$ refers to each neighbor and takes values from 0 to $P-1$.

- The function $s(x)$ acts as a binary threshold, defined as:

$$s(x) = \begin{cases} 1, & \text{if } x \geq 0, \\ 0, & \text{if } x < 0. \end{cases}$$

- Each neighbor is assigned a weight based on its position using the factor $2^p$.

As described in Equation (1), the LBP value encodes the intensity differences into a binary pattern, providing an efficient representation of the texture characteristics within the image.

**Histogram of oriented gradients.** The HOG feature descriptor is commonly deployed in object detection, especially in discovering the intensity gradients and their directions in smaller parts of an image [26]. HOG represents objects well by encoding the orientation of gradients in terms of histograms which can be used in applications including detecting pedestrians and vehicles.

The image is divided into small cells in order to derive the HOG descriptor. In every cell, the magnitudes and direction of the gradients are calculated as an estimate of each pixel. The magnitude of the gradient can be defined as Equation (2) where $G(x, y)$ is the gradient magnitude.

$$G(x, y) = \sqrt{\left(\frac{\partial g(x, y)}{\partial x}\right)^2 + \left(\frac{\partial g(x, y)}{\partial y}\right)^2}$$

(2)

where $\frac{\partial g(x,y)}{\partial x}$ and $\frac{\partial g(x,y)}{\partial y}$ represent the horizontal and vertical gradients of the image intensity $g(x,y)$, respectively. The gradient orientation, $\Theta(x, y)$, is then computed using Equation (3):

$$\Theta(x, y) = \tan^{-1}\left(\frac{\frac{\partial g(x,y)}{\partial y}}{\frac{\partial g(x,y)}{\partial x}}\right)$$

(3)

The pixels-per-cell parameter specifies the size of a cell in pixels (usually (8, 8)), and should be large enough to encode local gradient information. In each cell an estimation of the frequency distribution, weighted by the orientation magnitudes, is plotted as a histogram.

In order to be robust to minor changes in the image transformations and changes in illumination the histograms of the different cells are combined in blocks. These blocks are defined by the size parameter cells per block which is typically (2, 2). The normalization is done in an overlapping blocks and the increased invariance of the descriptor to lighting and contrast changes. The outcome of this process is a small and distinguishing vector of features and thus HOG is an excellent technique when handling visual tasks like object detection and shape recognition.

**Wavelet transform.**  Wavelet transform can be an efficient method to decompose image into different frequencies and scales with both space and frequency information and this can be used to capture fine details and smooth patterns in images [27]. It is well used in texture analysis, edge detection and other image processing tasks.

The wavelet transform breaks down an image into sub bands in terms of approximation and detail. Mathematically, DWT decomposition of an image is expressed as is displayed in Equation (4):

$$w(x, y) = \sum_{s=0}^{S-1} \sum_{t=0}^{T-1} \left(\lambda_{s,t} \cdot \psi_{s,t}(x, y)\right)$$

(4)

where:

- The function $w(x, y)$ represents the reconstructed image in the spatial domain.

- $\lambda_{s,t}$ are the wavelet coefficients, capturing the contribution of specific frequency components.

- $\psi_{s,t}(x, y)$ are the wavelet basis functions, where $s$ controls the scale (level of detail) and $t$ controls the position (translation).

- $S$ and $T$ denote the total number of scales and translations considered.

The wavelet that we used in our analysis was the db1 which is associated with an alternative of performing a four level decomposition on a specific image, that is, the wavelet-based image decomposition. This hierarchical structure allows extracting features at various levels thus it is very appropriate in carrying out activities in which texture analysis and edge detection are utilized. Table 1 displays the training/testing set dimension with respect to LBP, Wavelet and HOG feature on three data sets, by highlighting the variances of features extraction results.

## Classification

The classifiers that we used in this work are SVC, KNN, and Random Forest with a chosen set of parameters. The SVC [28] utilizes a linear kernel probability estimates with a set value of True (`kernel='linear', probability=True`) to decide the best decision boundaries that would be used in separation of classes. KNN [29] classifier employs 3 neighbors in nearest neighborhood (`n_neighbors=3`) whereby each data point is classified based on majority class of their neighbors. Random Forest [30] classifier employs 100 decision trees ((`n_estimators=100`) to enhance accuracy by combining the results of decision trees constructed on randomly chosen data and feature subsets.

In order to choose these best parameters we experimented with several hyperparameter values. In the case of KNN, we used $n\_neighbors = 1, 3, 5, 7, 9$; with random forest, we used $n\_estimators = 50, 100, 200$, $max\_depth = None, 10, 20$, and $min\_samples\_split = 2, 5$; and with SVC, we tested kernel (linear and rbf) and regularization ($C = 0.1, 1, 10$) and

**Table 1. Training and testing feature dimensions across datasets.**

| Dataset | Feature Type | Training | Testing |
|---|---|---|---|
| Dataset I | LBP | (6362, 16384) | (1311, 16384) |
| | Wavelet | (6362, 16384) | (1311, 16384) |
| | HOG | (6362, 8100) | (1311, 8100) |
| Dataset II | LBP | (3411, 16384) | (613, 16384) |
| | Wavelet | (3411, 16384) | (613, 16384) |
| | HOG | (3411, 8100) | (613, 8100) |
| Dataset III | LBP | (2400, 16384) | (600, 16384) |
| | Wavelet | (2400, 16384) | (600, 16384) |
| | HOG | (2400, 8100) | (600, 8100) |

gamma (scale, 0.01, 0.001). Three different datasets were experimented with, and the chosen parameters, namely KNN ($n\_neighbors = 3$), Random Forest ($n\_estimators = 100$), and Linear SVC, were regularly performing well across all datasets which makes up the final configuration employed in the given study.

## Probabilistic score fusion

The probabilistic score fusion model is the combination of various feature extraction algorithms, and classifiers to provide a better classification accuracy as shown in Algorithm 1. It aims to combine predictions made by various features and classifiers in order to enhance the efficiency of BTC. The approach critically examines the combinations of the classifiers residing on the various features, and determines its efficacy on the basis of the accuracy of fusion. It first examines scenarios when one classifier is used across all the features (the number of combinations is $m$, where $m$ denotes the number of total classifiers as well).

## Algorithm 1 LHW-Net Model for Brain Tumor Classification

**Require:** Preprocessed Images $\mathcal{I}$, Class Labels $\mathcal{L}$, Feature Extractors $\mathcal{F}$ (LBP, HOG, Wavelets), Classifiers $\mathcal{C}$ (KNN, SVC, Random Forest)
**Ensure:** Fused Predictions $\mathcal{P}_{fusion}$
 **Step 1: Feature Extraction**
 1: **for** each feature extractor $f \in \mathcal{F}$ **do**
 2: Extract training features: $\mathcal{F}_{train}[f] \leftarrow f(\mathcal{I}_{train})$
 3: Extract test features: $\mathcal{F}_{test}[f] \leftarrow f(\mathcal{I}_{test})$
 4: **end for**
 **Step 2: Classifier Training and Prediction**
 5: **for** each classifier $c \in \mathcal{C}$ **do**
 6: **for** each feature extractor $f \in \mathcal{F}$ **do**
 7: Train classifier: $c[f] \leftarrow c(\mathcal{F}_{train}[f], \mathcal{L}_{train})$
 8: Predict probabilities: $\mathcal{P}[f, c] \leftarrow c[f](\mathcal{F}_{test}[f])$
 9: **end for**
10: **end for**
 **Step 3: Probability Fusion**
11: **for** each combination of feature-classifier pairs $(f, c) \in \mathcal{F} \times \mathcal{C}$ **do**
12: Combine probabilities:
$$\mathcal{P}_{combined} \leftarrow \prod_{(f,c)} \mathcal{P}[f, c]$$
13: **end for**
14: Final predictions: $\mathcal{P}_{fusion} \leftarrow \mathrm{argmax}(\mathcal{P}_{combined})$
 **Step 4: Performance Evaluation**
15: Calculate the performance measures, including Accuracy, Sensitivity, Precision, and F1-Score values.
 **return** $\mathcal{P}_{fusion}$

The model then goes over all the combinations of classifiers in all the features to analyze all possible situations with various classifiers. Take as an example that every image in the dataset is evaluated using 27 combinations of two features and 27 combinations of three features. Such an extensive study on feature-classifier combinations guarantees a strong and desirable analysis providing overall improvements in the classification.

The total number of classifier-feature combinations for two and three features is given by:

$$\text{Total Combinations} = \binom{n}{i} \times m^i \tag{5}$$

where $n$ is the total number of features available, $i$ is the number of features selected, and $m$ is the number of classifiers. The term $\binom{n}{i}$ represents the number of ways to select $i$ features from $n$, and $m^i$ accounts for assigning one of $m$ classifiers to each selected feature.

For example, in the case of two features ($i=2$), the total combinations are:

$$\text{Total Combinations for Two Features} = \binom{n}{2} \times m^2 \tag{6}$$

Similarly, for three features ($i=3$), the total combinations are:

$$\text{Total Combinations for Three Features} = \binom{n}{3} \times m^3 \tag{7}$$

The formula is able to exhaustively check all the feature-classifier combinations, as indicated in Equation (5). In particular instances, Equations (6) and (7) outline the number of all possible combinations with 2 and 3 features respectively, promoting the robust and comprehensive study. In every combination, one uses precomputed probabilities given features based on individual feature-classifiers. These probabilities are then combined by multiplying them together across features and then the overall class prediction is chosen by getting the one with the highest overall probability. The fusion accuracy is computed in all combinations and the configuration that returns the best accuracy is determined as the best combination. This will result in effective performance of robust and accurate classification. The process is outlined as follows:

1. **Feature-Classifier Prediction:** Each feature extraction technique is paired with a classifier to generate class probabilities for the test dataset.

2. **Probability Fusion:** The predictions from all feature-classifier pairs are aggregated using a probabilistic fusion approach, as shown in Equation (8).

$$P_{\text{combined}} = \prod_{(f,c)} P[f, c] \tag{8}$$

where $P[f, c]$ represents the predicted probabilities for a specific feature-classifier pair.

3. **Final Prediction:** The final class label is assigned based on the maximum value of the combined probabilities, as described in Equation (9).

$$P_{\text{fusion}} = \text{argmax}(P_{\text{combined}}) \tag{9}$$

The strength of various feature extraction techniques and classifiers are leveraged in such a way that the classification model is strong and accurate due to this fusion methodology. Through the integration of predictions of individual features, two-feature combinations and three-feature combinations, the proposed model LHW-Net eliminates the factors that limit the functioning of individual models and also their inability to improve optimal predictive performance.

### Performance measures

The performance of the models was measured by four important parameters accuracy, precision, sensitivity and F1-score. Such metrics were computed against each feature extraction method in combination with different classifiers. Also, the LHW-Net performance was analyzed where multiple classifier combinations were applied. The in-depth review indicates the performance of individual characteristics and performance improvement achieved when the characteristics were integrated via fusion of classifiers. The underlying performance metrics are as follows:

- **Accuracy**: The correct classification of the instances with regard to total samples.

- **Precision**: This is the ratio of the correctly predicted positive to the total number predicted positive cases.

- **Sensitivity**: Also known as recall, and it is an indicator of the effectiveness of the model to identify all the positive cases.

- **F1 score**: It is an integrated measure that gives a compromise of precision and recall.

### Experimental results and discussion

The experiments were carried out on Linux OS with x86_64 architecture, 504 GB of RAM, and an NVIDIA Tesla V100-SXM2–32GB GPU. The system was using NVIDIA driver version 555.42.02, and it supports CUDA runtime up to 12.5. Nonetheless, our experiments were performed on the Python version 3.8.5 and TensorFlow 2.4.0, which was compiled against CUDA 11.2 and cuDNN 8 and thus the computations were done under this setup.

### Ablation study for base learner identification

The ablation study aims at determining the best possible combination of base learners and feature sets that can lead to better performance of a set of ensembles. On Dataset I, single feature–single classifier combination tests were conducted on single LBP, HOG, and Wavelets features among others to get the baseline accuracies of individual classifiers, which are SVC, KNN, and Random Forest. The paper has also examined combinations of features on a pair-wise basis (e.g., LBP + HOG, HOG + Wavelet) and three-way combinations (LBP + HOG + Wavelet) using a probabilistic method to fuse the scores to determine their combined effect. This analysis also gave insights into which features and classifier contributed the most, thus it was possible to identify combinations that could result in a much more accurate and robust ensemble of BTC.

Table 2 shows the classification results attained using one feature at a time on Dataset I. Having tested the combinations, the results turned out that two-feature combinations were more accurate than the individual features. Moreover, the combination of three features gave a bit more accurate results compared to the two-features combinations, which confirms that multiple features are more beneficial to be combined. The results of the two-feature combinations and the three-feature combinations, as analyzed by probabilistic score fusion method, have been tabulated in Table 3.

The comparison of ablation study notes that we achieved a good accuracy of using complementary features and classifiers as can be seen in the Fig 6. The working procedure is initiated by a test image, on which three features are obtained including LBP, HOG, and Wavelet Transform. These features are handled separately by their particular classifier, i.e., LBP with SVC, HOG with KNN, and Wavelet Transform with SVC. The results of these classifiers are then fused using probabilistic score fusion method in coming up with the final class label which takes a value between 1 to k (where k is

**Table 2. Individual feature results for Dataset I.**

| Feature | Classifier | Train Accuracy (%) | Test Accuracy (%) | Precision (%) | Sensitivity (%) | F1 Score (%) |
|---------|-----------|-------------------|-------------------|---------------|-----------------|--------------|
| LBP | KNN | 84.58 | 67.35 | 75.18 | 67.35 | 66.59 |
| LBP | RF | 99.95 | 89.55 | 89.78 | 89.55 | 89.53 |
| LBP | SVC | 99.95 | 90.31 | 90.25 | 90.31 | 90.23 |
| HOG | KNN | 96.76 | 93.36 | 93.54 | 93.36 | 93.19 |
| HOG | RF | 99.95 | 90.92 | 91.38 | 90.92 | 90.80 |
| HOG | SVC | 99.95 | **93.67** | 93.65 | 93.67 | 93.63 |
| Wavelet | KNN | 94.26 | 89.93 | 90.03 | 89.93 | 89.59 |
| Wavelet | RF | 99.95 | 92.30 | 92.49 | 92.30 | 92.25 |
| Wavelet | SVC | 99.95 | 90.31 | 90.25 | 90.31 | 90.28 |

the number of classes). The method has the advantage of taking a benefit of advantages of both feature-classifier pair, leading to better accuracy and robustness as is observed in the experimental data.

## Individual feature experimental results

This part represents an experimental part with emphasis on performance of single features (LBP, HOG, and Wavelet) classification in two more datasets. Several classifiers were used to determine the best features-classifier choices by dataset. Also, the multi-class and binary classification tasks were performed on a three-class dataset and a binary dataset. The performance of various features and models in term of the classification accuracies on Dataset II and Dataset III are summarized in Tables 4 and 5 respectively. The maximum accuracies of each of the datasets are depicted in the tables.

The findings show that feature-classifier pairs are different in their performance based on the type of dataset, which gives an idea of a configuration that reaches a higher accuracy rate and robustness. HOG always outperformed other features, and this only highlights its effectiveness in the representation of texture details, which are important in making accurate BTC. Different datasets gave different results regarding classifiers with SVC and KNN being among the most frequently accurate.

## Probability score fusion experimental results

In this section, the performance of the classification is demonstrated by having predictions of several pairs of the features-classifiers combined by employing the newly suggested LHW-Net strategy. This technique significantly increased classification quality on most datasets, which is an indication of the effectiveness of production of the complementary advantages of different features and different classifiers. The results for each dataset are represented in the following figures.

Regarding each dataset, the results, obtained in terms of confusion matrices, of both methods using individual sets of features and the fusion technique are presented visually in Figs 7–9. The given matrices offer some comparative analysis, as they depict the differences between various feature extraction techniques and their combination with regard to their performance in relation to the given datasets. Fig 10 shows the result of comparing individual feature-based and fusion results of each dataset. The suggested LHW-Net algorithm is much more accurate on the classification. As two features are combined, the accuracy increases by contrast to single features, and integrating three features brings a slight increase in accuracy.

Table 6 contains the overall comparison in terms of the accuracy, sensitivity, precision and F1 score of the models on the three datasets and allows a more accurate comparison in terms of numerical values. On all the datasets, the suggested model demonstrates better results in all the metrics of evaluation. This proves to show how strong the

**Table 3.** Two-feature and three-feature combination results for Dataset I.

| Feature Combination | Accuracy (%) | Precision (%) | Sensitivity (%) | F1 Score (%) |
|---|---|---|---|---|
| {LBP: KNN, HOG: KNN} | 90.31 | 91.47 | 90.31 | 90.44 |
| {LBP: KNN, HOG: RF} | 85.89 | 88.65 | 85.89 | 86.18 |
| {LBP: KNN, HOG: SVC} | 87.95 | 90.16 | 87.95 | 88.26 |
| {LBP: RF, HOG: KNN} | 95.96 | 95.99 | 95.96 | 95.94 |
| {LBP: RF, HOG: RF} | 91.46 | 91.80 | 91.46 | 91.37 |
| {LBP: RF, HOG: SVC} | 93.52 | 93.60 | 93.52 | 93.47 |
| {LBP: SVC, HOG: KNN} | 96.11 | 96.15 | 96.11 | 96.10 |
| {LBP: SVC, HOG: RF} | 90.85 | 90.94 | 90.85 | 90.74 |
| {LBP: SVC, HOG: SVC} | 93.67 | 93.72 | 93.67 | 93.62 |
| {LBP: KNN, Wavelet: KNN} | 87.41 | 89.41 | 87.41 | 87.77 |
| {LBP: KNN, Wavelet: RF} | 86.96 | 89.42 | 86.96 | 87.30 |
| {LBP: KNN, Wavelet: SVC} | 86.42 | 88.01 | 86.42 | 86.74 |
| {LBP: RF, Wavelet: KNN} | 94.51 | 94.44 | 94.51 | 94.45 |
| {LBP: RF, Wavelet: RF} | 91.83 | 92.02 | 91.83 | 91.76 |
| {LBP: RF, Wavelet: SVC} | 92.67 | 92.74 | 92.67 | 92.66 |
| {LBP: SVC, Wavelet: KNN} | 94.58 | 94.52 | 94.58 | 94.54 |
| {LBP: SVC, Wavelet: RF} | 91.22 | 91.25 | 91.22 | 91.16 |
| {LBP: SVC, Wavelet: SVC} | 92.83 | 92.77 | 92.83 | 92.78 |
| {HOG: KNN, Wavelet: KNN} | 94.51 | 94.67 | 94.51 | 94.37 |
| {HOG: KNN, Wavelet: RF} | **96.34** | 96.35 | 96.34 | 96.32 |
| {HOG: KNN, Wavelet: SVC} | 96.10 | 96.09 | 96.10 | 96.09 |
| {HOG: RF, Wavelet: KNN} | 94.81 | 94.76 | 94.81 | 94.76 |
| {HOG: RF, Wavelet: RF} | 91.68 | 91.95 | 91.68 | 91.60 |
| {HOG: RF, Wavelet: SVC} | 93.44 | 93.45 | 93.44 | 93.40 |
| {HOG: SVC, Wavelet: KNN} | 96.11 | 96.08 | 96.11 | 96.08 |
| {HOG: SVC, Wavelet: RF} | 94.20 | 94.30 | 94.20 | 94.16 |
| {HOG: SVC, Wavelet: SVC} | 94.58 | 94.51 | 94.58 | 94.54 |
| {LBP: KNN, HOG: KNN, Wavelet: KNN} | 90.39 | 91.84 | 90.39 | 90.58 |
| {LBP: KNN, HOG: KNN, Wavelet: RF} | 90.39 | 91.47 | 90.39 | 90.50 |
| {LBP: KNN, HOG: KNN, Wavelet: SVC} | 91.08 | 92.27 | 91.08 | 91.19 |
| {LBP: KNN, HOG: RF, Wavelet: KNN} | 89.78 | 90.85 | 89.78 | 89.92 |
| {LBP: KNN, HOG: RF, Wavelet: RF} | 87.26 | 89.60 | 87.26 | 87.55 |
| {LBP: KNN, HOG: RF, Wavelet: SVC} | 87.26 | 89.14 | 87.26 | 87.56 |
| {LBP: KNN, HOG: SVC, Wavelet: KNN} | 90.24 | 91.41 | 90.24 | 90.38 |
| {LBP: KNN, HOG: SVC, Wavelet: RF} | 88.33 | 90.44 | 88.33 | 88.61 |
| {LBP: KNN, HOG: SVC, Wavelet: SVC} | 88.48 | 90.27 | 88.48 | 88.75 |
| {LBP: RF, HOG: KNN, Wavelet: KNN} | 96.03 | 96.01 | 96.03 | 96.01 |
| {LBP: RF, HOG: KNN, Wavelet: RF} | 96.11 | 96.17 | 96.11 | 96.10 |
| {LBP: RF, HOG: KNN, Wavelet: SVC} | 96.42 | 96.42 | 96.42 | 96.40 |
| {LBP: RF, HOG: RF, Wavelet: KNN} | 94.66 | 94.62 | 94.66 | 94.61 |
| {LBP: RF, HOG: RF, Wavelet: RF} | 91.69 | 91.97 | 91.69 | 91.60 |
| {LBP: RF, HOG: RF, Wavelet: SVC} | 92.75 | 92.91 | 92.75 | 92.71 |
| {LBP: RF, HOG: SVC, Wavelet: KNN} | 95.88 | 95.84 | 95.88 | 95.85 |
| {LBP: RF, HOG: SVC, Wavelet: RF} | 93.67 | 93.78 | 93.67 | 93.62 |
| {LBP: RF, HOG: SVC, Wavelet: SVC} | 93.97 | 93.98 | 93.97 | 93.93 |

*(Continued)*

| Feature Combination | Accuracy (%) | Precision (%) | Sensitivity (%) | F1 Score (%) |
|---|---|---|---|---|
| {LBP: SVC, HOG: KNN, Wavelet: KNN} | 96.34 | 96.33 | 96.34 | 96.33 |
| {LBP: SVC, HOG: KNN, Wavelet: RF} | 95.96 | 95.97 | 95.96 | 95.94 |
| {LBP: SVC, HOG: KNN, Wavelet: SVC} | **96.49** | 96.48 | 96.49 | 96.48 |
| {LBP: SVC, HOG: RF, Wavelet: KNN} | 94.58 | 94.53 | 94.58 | 94.54 |
| {LBP: SVC, HOG: RF, Wavelet: RF} | 91.61 | 91.78 | 91.61 | 91.52 |
| {LBP: SVC, HOG: RF, Wavelet: SVC} | 92.98 | 93.01 | 92.98 | 92.93 |
| {LBP: SVC, HOG: SVC, Wavelet: KNN} | 95.42 | 95.38 | 95.42 | 95.39 |
| {LBP: SVC, HOG: SVC, Wavelet: RF} | 93.52 | 93.57 | 93.52 | 93.47 |
| {LBP: SVC, HOG: SVC, Wavelet: SVC} | 94.43 | 94.44 | 94.43 | 94.40 |

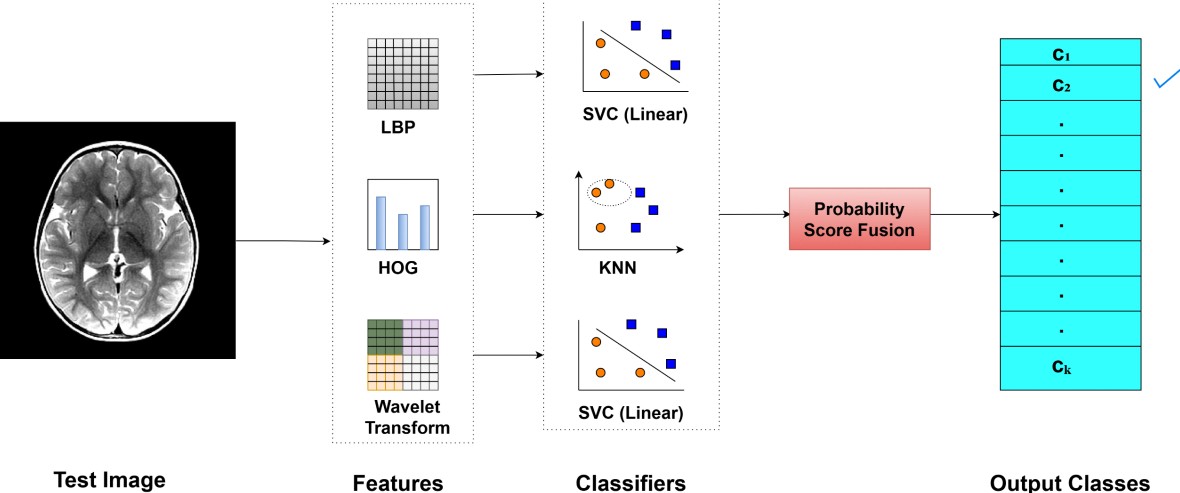

**Fig 6. Improved accuracy through feature extraction and classifier fusion using LBP, HOG, and Wavelet Transform.**

**Table 4. Individual feature results for Dataset II.**

| Feature | Classifier | Train Accuracy (%) | Test Accuracy (%) | Precision (%) | Sensitivity (%) | F1 Score (%) |
|---|---|---|---|---|---|---|
| LBP | KNN | 60.48 | 47.80 | 28.29 | 47.80 | 34.91 |
| LBP | RF | 99.97 | 82.22 | 84.65 | 82.22 | 82.73 |
| LBP | SVC | 99.97 | 89.40 | 89.59 | 89.40 | 89.43 |
| HOG | KNN | 98.24 | **95.27** | 95.24 | 95.27 | 95.24 |
| HOG | RF | 99.97 | 89.72 | 90.45 | 89.72 | 89.82 |
| HOG | SVC | 99.97 | 93.47 | 93.41 | 93.47 | 93.43 |
| Wavelet | KNN | 94.37 | 88.74 | 88.52 | 88.74 | 88.58 |
| Wavelet | RF | 99.97 | 90.70 | 91.46 | 90.70 | 90.84 |
| Wavelet | SVC | 99.97 | 87.44 | 87.50 | 87.44 | 87.45 |

**Table 5. Individual feature results for Dataset III.**

| Feature | Classifier | Train Accuracy (%) | Test Accuracy (%) | Precision (%) | Sensitivity (%) | F1 Score (%) |
|---------|-----------|--------------------|-------------------|---------------|-----------------|--------------|
| LBP | KNN | 97.50 | 94.83 | 94.86 | 94.83 | 94.83 |
| LBP | RF | 100.00 | 90.50 | 90.74 | 90.50 | 90.49 |
| LBP | SVC | 100.00 | 95.67 | 95.67 | 95.67 | 95.67 |
| HOG | KNN | 98.67 | **98.00** | 98.00 | 98.00 | 98.00 |
| HOG | RF | 100.00 | 91.50 | 91.54 | 91.50 | 91.50 |
| HOG | SVC | 100.00 | 97.33 | 97.33 | 97.33 | 97.33 |
| Wavelet | KNN | 97.54 | 90.33 | 91.49 | 90.33 | 90.27 |
| Wavelet | RF | 100.00 | 94.50 | 94.54 | 94.50 | 94.50 |
| Wavelet | SVC | 100.00 | 97.67 | 97.67 | 97.67 | 97.67 |

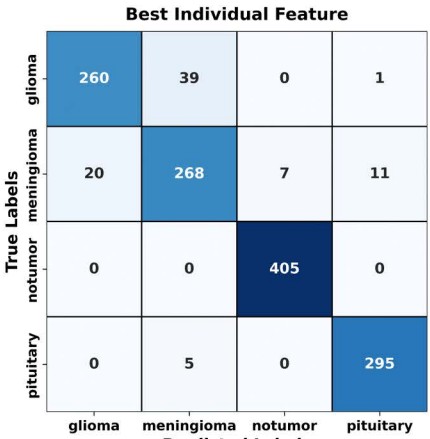
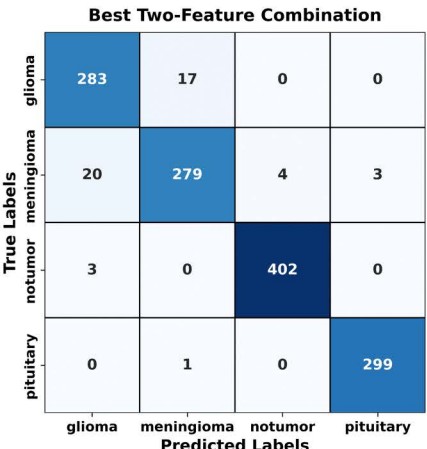
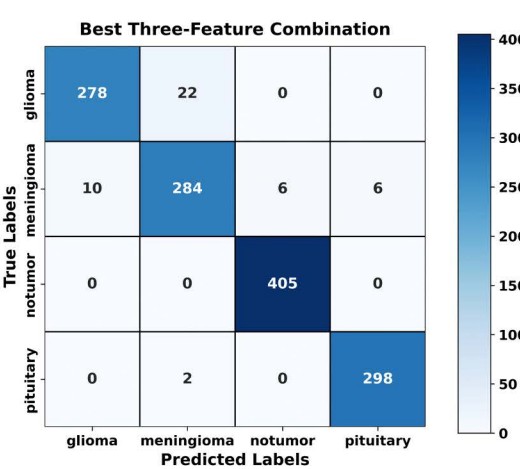

**Fig 7. Confusion matrix for Dataset I.**

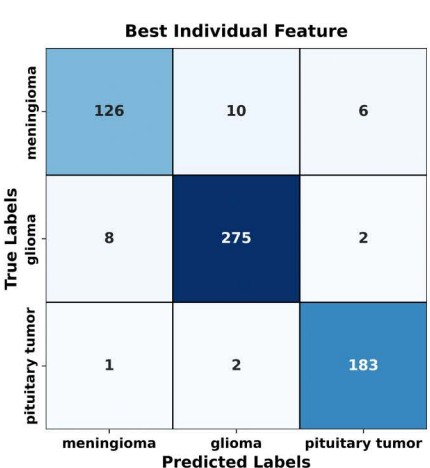
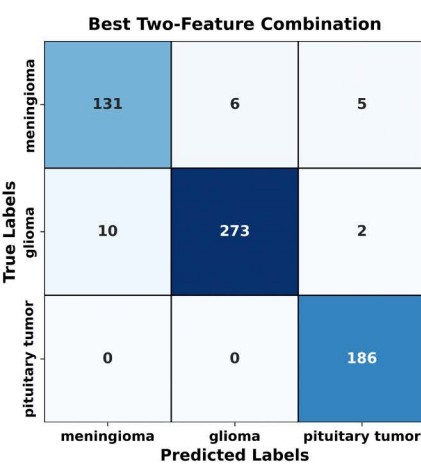
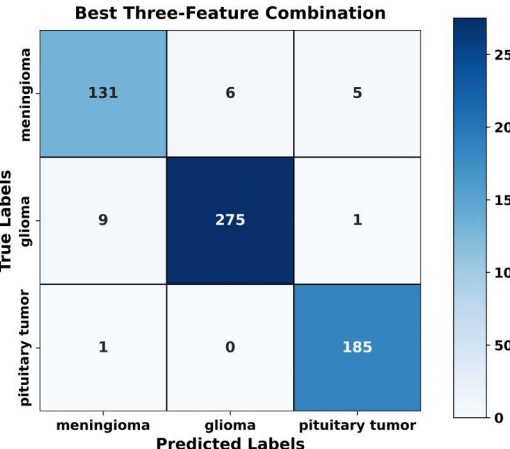

**Fig 8. Confusion matrix for Dataset II.**

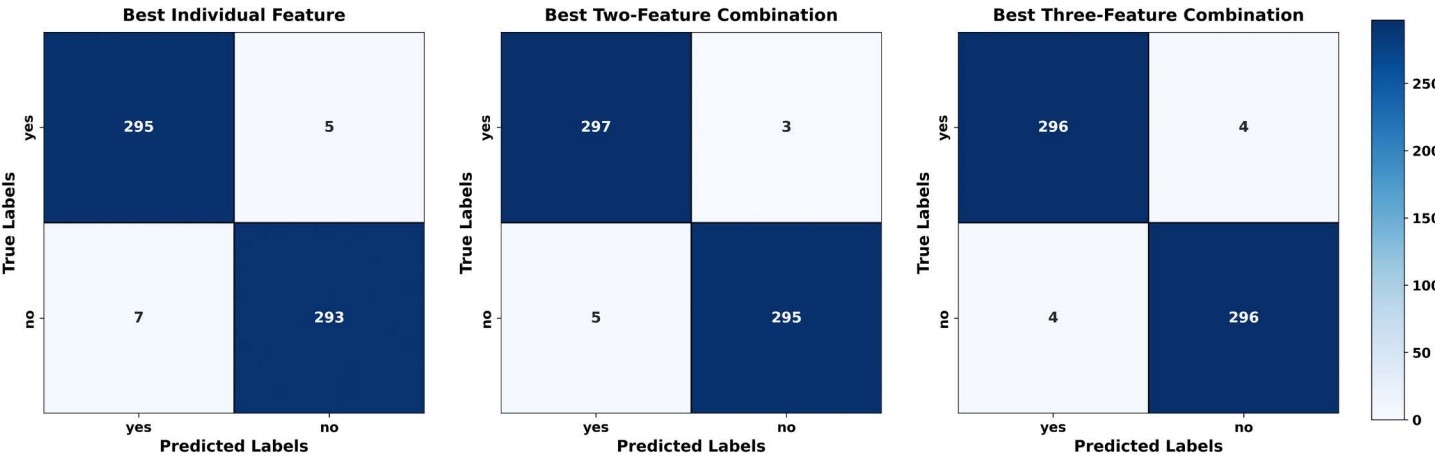

**Fig 9. Confusion matrix for Dataset III.**

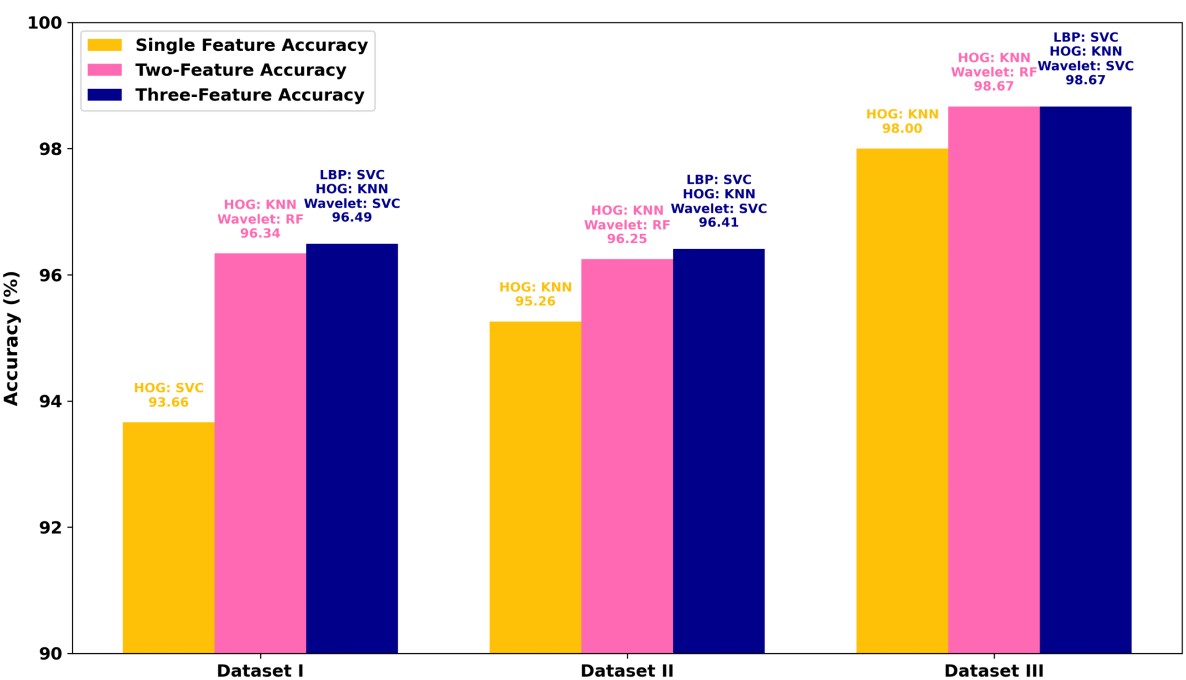

**Fig 10. Accuracy comparison of the best individual feature-based models and feature fusion models across datasets.**

methodology of fusion is to improve the accuracy and reliability of classification of various datasets. These findings are also supported by the tabular representation as it provides a systematic comparison of the proposed fusion with the improved results in an understandable and clear form.

In addition, the outstanding performance of the LHW-Net method can also be observed not only in the achievement parameters of accuracy but also in the greater values of precisions, sensitivities and F1 scores in all three datasets. This enhancement proves that the use of complementarity of the strength of specific feature-classifier pairings LBP with SVC, HOG with KNN, and Wavelet with SVC coupled by probabilistic score fusion makes reliable and generalized classification

**Table 6. Comparison of metrics for individual features, two-feature combination, and three-feature combination.**

| Dataset | Feature Type | Accuracy (%) | Precision (%) | Sensitivity (%) | F1 Score (%) |
|---|---|---|---|---|---|
| Dataset I | Individual | 93.66 | 93.65 | 93.66 | 93.63 |
| | Two-Feature Combination | 96.34 | 96.35 | 96.33 | 96.33 |
| | Three-Feature Combination | 96.49 | 96.48 | 96.49 | 96.48 |
| Dataset II | Individual | 95.27 | 95.24 | 95.27 | 95.24 |
| | Two-Feature Combination | 96.25 | 96.26 | 96.25 | 96.24 |
| | Three-Feature Combination | 96.41 | 96.41 | 96.41 | 96.40 |
| Dataset III | Individual | 98.00 | 98.00 | 98.00 | 98.00 |
| | Two-Feature Combination | 98.67 | 98.67 | 98.67 | 98.67 |
| | Three-Feature Combination | 98.67 | 98.67 | 98.67 | 98.67 |

possible. These findings confirm that not only does the fusion strategy reduce the detriments of individual feature sets or classifiers, but it can also allow a more comprehensive decision-making paradigm. As a result, the proposed model has great promise in classifying brain tumors based on work under different data distributions, hence proving its competence to deal with most real-life complex problems with greater accuracy and stability.

A comparison of the proposed technique with some of the more established approaches is given in Table 7. It provides a gamut of approaches that entails hand-made features like Gabor filters, HOG, and DSURF, and deep learning handled approaches like VGG16, ResNet, DenseNet, CNN, and Siamese Networks. Some of the hybrid approaches which

**Table 7. Comparison of Classification approaches across all datasets with existing literature.**

| Author | Year | Dataset (Split) | Method | Accuracy (%) |
|---|---|---|---|---|
| [31] | 2023 | Dataset I | Gabor filter and ResNet50 features were extracted from MRI images and classified individually and combined using SVM | 95.73 |
| [32] | 2024 | Dataset I | VGG16, ResNet18, and DenseNet pretrained models were compared for BTC | 95.00 |
| [33] | 2023 | Dataset I | HOG + Machine Learning Classifiers | 92.02 |
| [34] | 2026 | Dataset I | BoT-YOLOv6 | 96.37 |
| [35] | 2024 | Dataset I | Combination of SVM, HOG, LBP, and PCA | 96.03 |
| [36] | 2024 | Dataset I | Vision Transformer with self-attention, relative positional encoding, and residual MLP | 91.36 |
| **Proposed Model** | | Dataset I | Combination of Feature Fusion (LBP + SVC, HOG + KNN, Wavelet+SVC) | **96.49** |
| [1] | 2022 | Dataset II | An 8-layer classical CNN-based network was developed for classification | 93.83 |
| [37] | 2021 | Dataset II | Siamese Neural Network | 92.60 |
| [10] | 2019 | Dataset II | A GAN-pretrained deep neural network was fine-tuned to classify tumor classes | 93.01(introduced split) |
| [38] | 2022 | Dataset II | Multi-Channel CNN | 89.81 |
| [14] | 2022 | Dataset II | Normalization, DSURF, and HOG features classified with SVM | 90.27 |
| [39] | 2024 | Dataset II | VGG16 | 95 |
| **Proposed Model** | | Dataset II | Combination of Feature Fusion (LBP + SVC, HOG + KNN, Wavelet+SVC) | **96.41** |
| [40] | 2023 | Dataset III (8:2) | CNN optimized using improved Political Optimizer | 97.09 |
| [41] | 2023 | Dataset III | A new CNN architecture for BTC | 97.20 |
| [1] | 2023 | Dataset III (8:2) | An 8-layer classical CNN-based network was developed for classification | 97.26 |
| [42] | 2025 | Dataset III | Optimized Custom-CNN | 97.22 |
| **Proposed Model** | | Dataset III | Combination of Feature Fusion (LBP + SVC, HOG + KNN, Wavelet+SVC) | **98.67** |

incorporate feature extraction algorithms with classifiers such as SVM have also been mentioned. The suggested model is based on the idea of feature-wise combination of LBP, HOG, and Wavelet features with multiple classifiers used to improve the performance. The proposed approach proves to be very robust, scalable, and accurate on all three datasets as indicated by the results. This assessment proves that it is better than the current approaches, which underlines its great usefulness and efficiency.

However, some limitations are to be considered in the proposed work. The framework is based on handcrafted features, which might not be able to capture deeper semantic information in MRI images. Also, the model training takes much time, especially the SVC classifier when using large feature sets. Future directions will be to solve these problems through the combination of handcrafted and deep learning-based features, domain adaptation mechanisms to enhance robustness to a variety of datasets, and classifier optimization to achieve accuracy with minimal training complexity.

## Conclusion

The current study focuses on brain tumour classification using a strong framework, which extracts handcrafted features based on LBP, HOG, and Wavelet transforms. These features are then combined by using a selection of machine learning classifiers. A probabilistic score fusion strategy allows combining the results of the individual classifiers and greatly increases the quality of the classification. An experimental study based on standard popular benchmark datasets proves the accuracy and efficiency of the proposed framework. Combining handcrafted features with ensemble strategies is also an area of potential explored in the present work.

## Supporting information

**S1 File. Description of the datasets used in this study, with download links.**
(PDF)

## Author contributions

**Conceptualization:** Thireesha Suryadevara, Naveenkumar Mahamkali.

**Data curation:** Naveenkumar Mahamkali.

**Formal analysis:** Thireesha Suryadevara, Naveenkumar Mahamkali.

**Funding acquisition:** Mudassir Rafi.

**Investigation:** Thireesha Suryadevara.

**Methodology:** Thireesha Suryadevara, Naveenkumar Mahamkali.

**Project administration:** Mudassir Rafi.

**Supervision:** Naveenkumar Mahamkali, Mudassir Rafi.

**Validation:** Naveenkumar Mahamkali.

**Visualization:** Naveenkumar Mahamkali.

**Writing – original draft:** Thireesha Suryadevara.

**Writing – review & editing:** Thireesha Suryadevara, Mudassir Rafi.

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
