## [Decision Letter · Decision Letter 0]

18 Sep 2025

Dear Dr. Rafi,

Thank you for submitting your manuscript to PLOS ONE. After careful consideration, we feel that it has merit but does not fully meet PLOS ONE’s publication criteria as it currently stands. Therefore, we invite you to submit a revised version of the manuscript that addresses the points raised during the review process.

We look forward to receiving your revised manuscript.

Kind regards,

Anwar P.P. Abdul Majeed

Academic Editor

PLOS ONE

Journal Requirements:

The authors extend their appreciation to the Deanship of Research and Graduate Studies at King Khalid University for funding this work through Large Research Project under grant number RGP2/588/46.

5. Please upload a new copy of Figures 4 and 6 as the detail is not clear. Please follow the link for more information: https://blogs.plos.org/plos/2019/06/looking-good-tips-for-creating-your-plos-figures-graphics/

Reviewers' comments:

Reviewer's Responses to Questions

**Comments to the Author**

1. Is the manuscript technically sound, and do the data support the conclusions?

Reviewer #1: Partly

2. Has the statistical analysis been performed appropriately and rigorously?

Reviewer #1: Yes

3. Have the authors made all data underlying the findings in their manuscript fully available?

Reviewer #1: Yes

4. Is the manuscript presented in an intelligible fashion and written in standard English?

Reviewer #1: Yes

Reviewer #1: 1. The quality of the figures is low. Please consider providing the images in higher resolution, as the information in the images is difficult to read.

2. Did the authors split the data into training and testing sets after data augmentation? I am concerned about potential data leakage if this approach was used.

3. Could you double check if this CUDA version is correct: CUDA 12.5555.42.02?

4. The part of "Ablation Study for Base Learner Identification" should be moved to the section of "Experimental Results and Discussion", as it presents the findings of your experiment.

5. The "Performance measures" should be moved to the section of "Materials and Methods", as it describes the evaluation metrics used to measure performance.

6. In Table 7, please specify the classifier that produced the optimal outcomes in your proposed model, rather than writing “Combination of Feature Fusion (LBP, HOG, Wavelet) with Various Classifiers.”

7. I am a bit confused about the design of the proposed model. Is it an ensemble model with three classifiers at the last layer, or is it three individual models, each with one classifier (KNN, RF, SVC)?

8. What are the limitations of the experiment? This should be mentioned in the section of "Experimental Results and Discussion".

.

Reviewer #1: **Yes:** Yun Xin TeohYun Xin TeohYun Xin TeohYun Xin Teoh

---

## [Author Response · Author response to Decision Letter 1]

22 Oct 2025

Dear Reviewer (s),

We sincerely thank you for your time and effort in reviewing our manuscript. Your detailed and constructive feedback has been invaluable in improving the clarity, rigor, and overall quality of the work. We have carefully addressed each of your comments and incorporated your suggestions into the revised version.

Reviewer 1:

Comment 1:

The quality of the figures is low. Please consider providing the images in higher resolution, as the information in the images is difficult to read. Response:

The figures have been replaced with more detailed ones (300 dpi) to make them clear and readable. Moreover, the subfigures are eliminated and rearranged as separate figures to present them better. The revised version of the manuscript consists of clearer plots, confusion matrices, and sample images to visualize them better.

Location in Paper: Throughout the paper.

Comment 2:

Did the authors split the data into training and testing sets after data augmentation? I am concerned about potential data leakage if this approach was used.

Response: In the revised version, we ensured that no data leakage occurred. For Dataset I, the data was already provided as predefined training and test sets, and only the training set was augmented. For Datasets II and III, the data was first divided into training and test sets, after which only the training sets were augmented.

The overall process of data splitting and augmentation has been clearly illustrated in Figures 3 and 5 of the revised manuscript.

Location in Paper: Page numbers: 05, Line number 180 and 181.

Comment 3:

Could you double check if this CUDA version is correct: CUDA 12.5555.42.02? Response: Thank you for your suggestion. The CUDA 12.5 version is supported by the driver, whereas our experiments were based on TensorFlow 2.4.0 and CUDA 11.2, cuDNN 8. Manuscript was updated to eliminate confusion.

Location in Paper: Page numbers: 11, Line number 345 to 350.

Comment 4:

The part of "Ablation Study for Base Learner Identification" should be moved to the section of "Experimental Results and Discussion", as it presents the findings of your experiment.

Response: As per your suggestion the manuscript had restructured accordingly.

Location in Paper: Page numbers: 11 and 12.

Comment 5:

The "Performance measures" should be moved to the section of "Materials and Methods", as it describes the evaluation metrics used to measure performance.

Response: As per your suggestion the manuscript had restructured accordingly.

Location in Paper: Page numbers: 10.

Comment 6:

In Table 7, please specify the classifier that produced the optimal outcomes in your proposed model, rather than writing “Combination of Feature Fusion (LBP, HOG, Wavelet) with Various Classifiers.”

Response: Table 7 has been updated to include a specific reference to the classifier(s) that obtained the best results on each dataset (e.g. LBP+SVC, HOG+KNN, Wavelet+SVC) .

Location in Paper: Page numbers: 15.

Comment 7: I am a bit confused about the design of the proposed model. Is it an ensemble model with three classifiers at the last layer, or is it three individual models, each with one classifier (KNN, RF, SVC)?

Response: In the initial stage, we employed three different classifiers—KNN, RF, and SVC—to train individually on distinct feature sets (LBP, HOG, and Wavelet, respectively). We then evaluated multiple feature–classifier pairings through probabilistic score fusion. The most effective combinations identified were LBP + SVC, HOG + KNN, and Wavelet + SVC, which were subsequently integrated at the decision level.

Therefore, the proposed approach does not consist of three independent models but represents a single ensemble framework, wherein feature-specific classifiers contribute to a unified final decision.

Location in Paper: This clarification has been added to the revised manuscript on Page 14, Lines 421–424.

Comment 8: What are the limitations of the experiment? This should be mentioned in the section of "Experimental Results and Discussion".

Response: In the revised manuscript, we have explicitly mentioned the limitations in the “Experimental Results and Discussion” section. The current framework demonstrates promising accuracy; however, it has the following limitations:

1. The proposed model relies on handcrafted feature descriptors, which may not capture deeper semantic representations inherent in complex attack patterns.

2. The approach is computationally intensive, particularly when using the SVC classifier with large feature sets, leading to increased processing time.

These limitations have been discussed along with potential directions for improvement in future work.

Location in Paper: The corresponding details are provided on Page 14, Lines 442–448 of the revised manuscript.

Response to Editorial Requirements:

Comment 1: Please ensure that your manuscript meets PLOS ONE's style requirements, including those for file naming. Response: We have modified the manuscript according to the PLOS ONE LaTeX template, to ensure that it is presented according to the style standards of the journal such as file names and formatting.

Comment 2: Please note that PLOS One has specific guidelines on code sharing for submissions in which author-generated code underpins the findings in the manuscript. In these cases, we expect all author-generated code to be made available without restrictions upon publication of the work. Response: The author-created code of this study is stored in the following GitHub repository: (https://github.com/thireesha-suryadevara/LHW-Net).

Comment 3: Thank you for stating the following financial disclosure:

The authors extend their appreciation to the Deanship of Research and Graduate Studies at King Khalid University for funding this work through Large Research Project under grant number RGP2/588/46.

Response: The author Dr. Mudassir Rafi is having dual affiliation as mentioned in the manuscript. Currently, he is associated with the funder, King Khalid University, as an Assistant Professor and as one of the supervisors for the first author, Mrs Thireesha Suryadevara. He is involved in Supervision, reviewing, advising and fund acquisition for the submitted work as stated in the financial disclosure.

Comment 4: When completing the data availability statement of the submission form, you indicated that you will make your data available on acceptance. We strongly recommend all authors decide on a data sharing plan before acceptance, as the process can be lengthy and hold up publication timelines. Please note that, though access restrictions are acceptable now, your entire data will need to be made freely accessible if your manuscript is accepted for publication. This policy applies to all data except where public deposition would breach compliance with the protocol approved by your research ethics board. If you are unable to adhere to our open data policy, please kindly revise your statement to explain your reasoning and we will seek the editor's input on an exemption. Please be assured that, once you have provided your new statement, the assessment of your exemption will not hold up the peer review process.

Response: We had updated the Data Availability Statement to reflect our intended plan for sharing the data. The data used in this study is publicly available on Kaggle.

All code created by the authors (programming code) has been placed in (https://github.com/thireesha-suryadevara/LHW-Net) githhub repository.

Comment 5: Please upload a new copy of Figures 4 and 6 as the detail is not clear. Response: Figures 4 and 6 are also revised to higher-resolution images (at least 300 dpi) and the subfigures have been replaced to individual full-resolution images, which improves quality and visibility of details. The workflow is represented in figures 3, 5, 7, 8, and 9 of the revised manuscript.

Comment 6: Please include captions for your Supporting Information files at the end of your manuscript, and update any in-text citations to match accordingly. Response: We included captions for all Supporting Information files at the end of the manuscript, while independently ensuring in-text citations matched the respective files.

Comment 7: If the reviewer comments include a recommendation to cite specific previously published works, please review and evaluate these publications to determine whether they are relevant and should be cited. There is no requirement to cite these works unless the editor has indicated otherwise. Response: We thank the editor for this clarification. Upon reviewing all reviewer comments, we found that no additional citations were specifically recommended for inclusion. Therefore, no new references have been added in this revision.

---

## [Decision Letter · Decision Letter 1]

20 Feb 2026

Dear Dr. Rafi,

plosone@plos.org. . . . A letter that responds to each point raised by the academic editor and reviewer(s). You should upload this letter as a separate file labeled 'Response to Reviewers'.A marked-up copy of your manuscript that highlights changes made to the original version. You should upload this as a separate file labeled 'Revised Manuscript with Track Changes'.An unmarked version of your revised paper without tracked changes. You should upload this as a separate file labeled 'Manuscript'.

We look forward to receiving your revised manuscript.

Kind regards,

Hikmat Ullah Khan, PhD (Computer Science)

Academic Editor

PLOS One

Journal Requirements:

Reviewers' comments:

Reviewer's Responses to Questions

**Comments to the Author**

Reviewer #1: All comments have been addressed

Reviewer #2: All comments have been addressed

2. Is the manuscript technically sound, and do the data support the conclusions?

Reviewer #1: Yes

Reviewer #2: Yes

3. Has the statistical analysis been performed appropriately and rigorously?

Reviewer #1: Yes

Reviewer #2: Yes

4. Have the authors made all data underlying the findings in their manuscript fully available?

Reviewer #1: Yes

Reviewer #2: Yes

5. Is the manuscript presented in an intelligible fashion and written in standard English?

Reviewer #1: Yes

Reviewer #2: Yes

Reviewer #1: Thank you for the effort of addressing the comments. However I still have some comments regarding the revised document.

- Please don’t use yellow as a highlight color for revisions. It is difficult to read.

- For the figures, I still think the quality of images should be enhanced.

Reviewer #2: Improve resolution and clarity of all figures and increase font size in diagrams and confusion matrices.

Clarify preprocessing steps (resizing, normalization, etc.).

Explain hyperparameter tuning strategy.

Fix minor typos and formatting inconsistencies.

.

Reviewer #1: **Yes:** Yun Xin TeohYun Xin TeohYun Xin TeohYun Xin Teoh

Reviewer #2: No

---

## [Author Response · Author response to Decision Letter 2]

10 Mar 2026

Comment 1:

Please don’t use yellow as a highlight color for revisions. It is difficult to read. Response:

In the revised manuscript, the yellow highlight has been changed to a more readable color (blue) to enhance clarity.

Location in Paper: Throughout the paper.

Comment 2:

For the figures, I still think the quality of images should be enhanced.

Response:

Thank you for the comment. The figures have been improved in terms of resolution, font sizes, and overall clarity to improve their readability. All figures were checked with NewGen ARTANALYSIS (NAAS tool) in order to match the requirements of the journal in terms of formatting and presentation.

Location in Paper: Throughout the paper.

Comment 1:

Improve resolution and clarity of all figures and increase font size in diagrams and confusion matrices. Response:

Every figure has been updated with the improvement of their resolution, the enlargement of fonts in the diagrams and confusion matrices, and the overall clarity to make them easier to read and present better. All figures were checked with NewGen ARTANALYSIS (NAAS tool) in order to match the requirements of the journal in terms of formatting and presentation.

Location in Paper: Throughout the paper.

Comment 2:

Clarify preprocessing steps (resizing, normalization, etc.). Response:

We have revised the paper to explain the preprocessing steps, such as Gray scale conversion, resizing etc.

Location in Paper: Page number: 05, Line number 154 to 187.

Comment 3:

Explain hyperparameter tuning strategy. Response:

We have provided the description of the hyperparameter tuning strategy in the revised manuscript. The three models, which are KNN, Random Forest, and SVM, were tested on three datasets with varying hyperparameter settings. As the performance varied with different parameter settings, the optimal hyperparameters were empirically determined based on the best overall performance across the datasets. The selected parameters are: KNN (n_neighbors=3), Random Forest, (n_estimators=100), and Linear SVM.

Location in Paper: Page number: 08, Line number 274 to 290.

Comment 4:

Fix minor typos and formatting inconsistencies. Response:

The paper has been carefully checked, and all minor typographical errors and formatting discrepancies have been removed.

Location in Paper: Throughout the paper.

Response to Editorial Requirements:

Comment 1:

Response:

The reviewer did not make any specific previously published works to be cited. But we have reviewed the literature attentively and included a few recent and relevant references to make the manuscript more powerful and justified.

Location in Paper: Page number: 15.

Comment 2:

Response:

The reference list has been carefully reviewed to make sure that every entry is complete and correct. No retracted articles are mentioned. Moreover, some recent and relevant publications were added to further justify and prove the proposed work. The changes in the reference list have been added to the revised manuscript.

Location in Paper: Page number: 15.

---

## [Decision Letter · Decision Letter 2]

24 Mar 2026

LHW-Net: An Ensemble-Based Machine Learning Framework for Brain Tumor Classification

PONE-D-25-43376R2

Dear Dr. Rafi,

We’re pleased to inform you that your manuscript has been judged scientifically suitable for publication and will be formally accepted for publication once it meets all outstanding technical requirements.

Kind regards,

Hikmat Ullah Khan, PhD (Computer Science)

Academic Editor

PLOS One

Additional Editor Comments (optional):

Reviewers' comments:

Reviewer's Responses to Questions

**Comments to the Author**

Reviewer #2: All comments have been addressed

2. Is the manuscript technically sound, and do the data support the conclusions?

Reviewer #2: Yes

3. Has the statistical analysis been performed appropriately and rigorously?

Reviewer #2: Yes

4. Have the authors made all data underlying the findings in their manuscript fully available?

Reviewer #2: Yes

5. Is the manuscript presented in an intelligible fashion and written in standard English?

Reviewer #2: Yes

Reviewer #2: The manuscript has improved significantly compared with the previous revision, and the authors have addressed the concerns raised in earlier review rounds. I only have a few minor editorial observations:

In the abstract, the statement "The classification of brain tumors is an unsolved problem" appears to be a very strong claim and could be softened.

The abstract and conclusion are somewhat similar in content. Ideally, the conclusion should briefly highlight the key results or main findings of the study.

.

Reviewer #2: No

---

## [Editor Report · Acceptance letter]

PONE-D-25-43376R2

PLOS One

Dear Dr. Rafi,

I'm pleased to inform you that your manuscript has been deemed suitable for publication in PLOS One. Congratulations! Your manuscript is now being handed over to our production team.

Kind regards,

on behalf of

Dr. Hikmat Ullah Khan

Academic Editor

PLOS One